# Investigating Psychological Impact after Receiving Genetic Risk Results—A Survey of Participants in a Population Genomic Screening Program

**DOI:** 10.3390/jpm12121943

**Published:** 2022-11-22

**Authors:** Cara Zayac McCormick, Kristen Dilzell Yu, Alicia Johns, Gemme Campbell-Salome, Miranda L. G. Hallquist, Amy C. Sturm, Adam H. Buchanan

**Affiliations:** 1Department of Genomic Health, Geisinger, Danville, PA 17822, USA; 2Department of Population Health Sciences, Geisinger, Danville, PA 17822, USA; 323andMe, Sunnyvale, CA 94086, USA

**Keywords:** genetic testing, genomics, genomic screening, precision health, MyCode, emotional response, psychological impact, patient outcomes

## Abstract

Genomic screening programs have potential to benefit individuals who may not be clinically ascertained, but little is known about the psychological impact of receiving genetic results in this setting. The current study sought to further the understanding of individuals’ psychological response to receiving an actionable genetic test result from genomic screening. Telephone surveys were conducted with patient-participants at 6 weeks and 6 months post genetic result disclosure between September 2019 and May 2021 and assessed emotional response to receiving results via the FACToR, PANAS, and decision regret scales. Overall, 354 (29.4%) study participants completed both surveys. Participants reported moderate positive emotions and low levels of negative emotions, uncertainty, privacy concern, and decision regret over time. There were significant decreases in negative emotions (*p* = 0.0004) and uncertainty (*p* = 0.0126) between time points on the FACToR scale. “Interested” was the highest scoring discrete emotion (T1 3.6, T2 3.3, scale 0–5) but was significantly lower at 6 months (<0.0001). Coupled with other benefits of genomic screening, these results of modest psychological impact waning over time adds support to clinical utility of population genomic screening programs. However, questions remain regarding how to elicit an emotional response that motivates behavior change without causing psychological harm.

## 1. Introduction

Population genomic screening can identify individuals at risk for disease who might not otherwise be ascertained by clinical criteria-based testing [1,2,3,4]. This approach has the potential to trigger proactive risk management and early disease detection or prevention, with the goal of decreasing population morbidity and mortality [5,6,7,8,9]. Decreasing costs, increasing numbers of population genomic screening programs, expanded testing panels offered by commercial labs, and ongoing initiatives such as the NIH *All of Us* Research Program will result in greater numbers of individuals receiving genomic screening results [10,11,12,13,14,15,16]. Initial data suggest that 1–2% of individuals tested through genomic screening programs have an actionable variant associated with conditions designated by the CDC as Tier 1 genomic applications (hereditary breast and ovarian cancer, Lynch syndrome, and familial hypercholesterolemia); higher yields have been reported when broader panels are utilized [13,17,18]. Furthermore, early analyses have suggested cost-effectiveness of genomic screening in certain populations, such as for hereditary breast and ovarian cancer syndrome in young women [19]. Despite potential promise, outstanding questions remain regarding the clinical utility of population genomic screening, necessitating additional research to guide implementation [20,21,22].

One critical piece of data needed to execute genomic screening programs successfully and responsibly is the experiences of individuals who receive actionable results, including how they respond to the result and what value they ascribe to it. While some have raised concerns of potential psychological harm in the setting of opportunistic screening [23,24], there have been limited studies on the experience of unselected populations undergoing genomic screening, with even less light shed on the experience of individuals with an actionable result in these cohorts. Early qualitative data from our group’s genomic screening program have suggested a manageable psychological response to an actionable result (likely pathogenic/pathogenic variants associated with increased disease risk for which there are effective interventions to mitigate risk) and highlighted the need for further study in a larger cohort over time [25]. Although Halverson et al. [26] and Lemke et al. [27] found higher levels of negative emotions in those with positive results compared to those with negative results (the majority of their cohorts), overall psychological distress was low. Additionally, there has been limited research on the specific emotions experienced by recipients of actionable results. In broader genetic testing and family health history contexts, previous research has found strong emotions motivate information seeking and medical decision making to adhere to recommended screenings [28,29]. Examining emotional response and adjustment to receiving actionable results has implications for patients’ well-being and medical decision making related to results, impacting clinical utility.

Here we seek to further the understanding of individuals’ psychological response to an actionable genetic test result from genomic screening. Participants in the Geisinger MyCode^®^ Community Health Initiative (MyCode) consent to receiving clinically actionable genomic results, providing an opportunity to guide future practice by understanding the experience and psychological outcomes of receiving results in a robust, unselected healthcare cohort [30,31]. The current study evaluates MyCode participants’ reactions over time to receiving genetic results. We conducted participant surveys at two time points post-disclosure, assessing emotional response and decision regret regarding receiving results. This study fills a need for more data on the impact of receiving actionable results from genomic screening in an unselected healthcare system cohort.

## 2. Materials and Methods

### 2.1. Setting

MyCode, an electronic health record (EHR)-linked biobank, has enrolled over 300,000 participants to date from throughout the Geisinger clinical enterprise [30]. Participation in the biobank includes exome sequencing and return of clinically actionable results to patient-participants [31,32] via the MyCode Genomic Screening and Counseling program. As of September 2022, nearly 3700 patient-participants have received a pathogenic or likely pathogenic genetic result from this program. The Genomic Screening and Counseling Registry allows for the administration of surveys to patient-participants who received a MyCode result. MyCode and the registry are approved by the Geisinger Institutional Review Board (IRB).

### 2.2. Participants

All MyCode patient-participants 18 years of age and older who received a genetic result between September 2019 and May 2021 were invited to participate in a telephone survey at 6 weeks (T1) and 6 months (T2) post result disclosure. Patient-participants who could not be reached for result disclosure were not eligible to participate. 

### 2.3. Procedures

All surveys were administered through a Computer Aided Telephone Interview (CATI) system. The CATI system ensured that survey questions maintained the exact standardized wording and followed logical skip patterns. Interviewers received written materials prior to the telephone administration period and training, which included a question-by-question overview, supervised practice administration, and instruction on strategies for addressing potential problems and ensuring accurate data collection. 

The survey was administered in English by research interviewers employed in Geisinger’s Survey Research and Recruitment Core (SRRC). At least 9 phone call attempts were made to complete an interview at primary and alternate phone numbers as listed in the patient-participants’ electronic health record (EHR), when available. The calls were attempted at various times of day and days of the week to maximize the likelihood of reaching potential respondents. Patient-participants received a $10 gift card for each survey completed.

Three additional data points were collected from patients’ EHRs: whether patients had a completed visit with a genetic counselor (completed GC visit), type of result (cardiovascular risk, cancer risk, or miscellaneous), and race and ethnicity. Data were identified and extracted from the EHR by a data analyst.

### 2.4. Measures

The current study focused on measures related to psychological and emotional responses of participants that were included on two broader surveys at two time points. Survey measures assessed here included demographic questions, questions related to health and quality of life, and personal and family awareness of the genetic risk. Multiple measures were administered to assess psychological and emotional responses. The Feelings About genomiC Testing Results (FACToR) scale consists of 12 items, broken into 4 subscales: negative emotions, positive feelings, uncertainty, and privacy concerns [33]. As all MyCode results are clinically actionable, one question on the uncertainty scale related to not having definite disease prevention guidelines was eliminated to minimize confusion and misinformation among our patient-participants; therefore our scale consisted of 11 items, and scoring was adjusted to reflect this. Summary scores were calculated for each subscale by adding individual items; reverse scoring was completed for the positive subscale before summing, following methods in the validation paper [33]. Positive subscale ranges from 0–16, negative subscale ranges from 0–12, and uncertainty and privacy subscales range from 0–8. Higher scores indicate greater psychological impairment [33]. To gain a more in-depth assessment of emotional responses, we also utilized the Positive and Negative Affect Schedule (PANAS) which consists of 10 positive emotions and 10 negative emotions [34]. Participants were asked to indicate how much of each discrete emotion was provoked when thinking about their MyCode result. Each subscale ranges from 0–50; the higher the number, the greater the positive or negative affect. Mean scores of each discrete emotion were calculated. Finally, we used a 5-item scale [35] to investigate decision regret for enrolling in MyCode and receiving a result via MyCode. Two of the items were reverse coded [35]; for each item, a higher number indicates more regret. Scores were converted to a 0–100 scale by subtracting 1 from each item, and multiplying by 25 [35]. Items were summed and averaged for a final score between 0–100. Higher scores on these scales indicated more psychological reaction, decision regret, and positive and negative affect.

### 2.5. Analyses

Descriptive statistics, including means, standard deviations, and medians for continuous variables, and frequencies and percentages for categorical variables, are presented. Bivariate analyses were conducted to determine any significant associations between covariates of interest (completed GC visit, age at result receipt, sex, highest education, and type of gene [gene group]) and continuous outcomes (FACToR subscales, PANAS emotions, including positive and negative affect scales, and decision regret scores) at 6 weeks and at 6 months using general linear models (GLM). Similarly, bivariate GLMs were used to assess time as a potentially significant predictor for continuous outcomes. For outcomes where time was found to be a significant predictor in the bivariate analysis, adjusted models were constructed using covariates that were found to have a significant relationship to that outcome at the alpha level 0.05 at 6 weeks or 6 months. Significant differences were determined at the alpha level of 0.05. Analyses were conducted in SAS Enterprise Guide Version 8.2 (SAS Institute Inc., Cary, NC, USA). 

## 3. Results

### 3.1. Overall Sample Characteristics

Of the 1208 individuals who received both the T1 and T2 surveys, 354 individuals completed both surveys, for a response rate of 29.3%. Participants’ characteristics are detailed in Table 1. The majority of participants were female (62.6%), white, and non-Hispanic (87.6%), had at least some college education (70.2%), were married or living with a partner (66.8%), and were privately insured (56.4%). Almost half of participants were working for pay, and there was a diverse range of household incomes. Mean age at result receipt was 57.6 years. The majority of respondents (78.1%) and their family members (63.3%) were previously unaware of the genetic result. Most participants reported good or very good health (M = 2.8) and quality of life (M = 2.3) at the most recent assessment (T2). Roughly half (54.1%) of participants completed a genetic counseling (GC) visit post-disclosure and prior to the T2 survey. Cancer was the predominant gene group (44%) among respondents, followed by cardiac and miscellaneous (32.8% and 22.9%, respectively).

### 3.2. FACToR Scale

#### 3.2.1. Positive Subscale

At 6 weeks, the mean FACToR positive subscale score for participants was 8.5, indicating neutral feelings. At 6 months, the mean positive subscale remained near the midpoint (8.9, Table 2). 

Older age was significantly associated with higher scores (greater functional impairment) at 6 weeks and 6 months, while being female was significantly associated with lower subscale scores (lesser functional impairment) at 6 weeks and 6 months. Having a family member who was previously told of the risk for the genetic condition was significantly associated with lower scores (lesser functional impairment) at 6 weeks. Type of result (cancer, cardiac, or miscellaneous) was significantly associated with subscale scores at 6 weeks, with individuals with cardiac results having higher scores (greater functional impairment) compared to those with cancer. This difference by gene group was no longer significant at 6 months (Table 3).

#### 3.2.2. Negative Subscale

At 6 weeks, mean negative subscale scores were low (M = 2.9), and there was a significant decrease in negative scores at 6 months (M = 2.0) compared to 6 weeks (Table 2). Being female was significantly associated with higher negative scores (more negative emotions) at 6 weeks and 6 months. Type of result was also significantly associated with negative subscale scores at both time points. Those with miscellaneous results had significantly lower scores (less negative emotions) compared to those with cancer results at 6 weeks, while at 6 months, individuals with cardiac or miscellaneous results had significantly lower scores (less negative emotions) than those with cancer results. At 6 weeks, patients who completed a genetic counseling visit had significantly higher negative subscale scores compared to those who did not, but this was no longer significant at 6 months. At 6 months, older age was significantly associated with lower negative subscale scores (Table 3). After adjusting the model for significant covariates (having a genetic counseling visit, age, sex, and gene group), time is still a significant predictor of the negative subscale score, where there was a significant decrease in score at 6 months (Table 2).

#### 3.2.3. Privacy Subscale

At 6 weeks, mean privacy subscale scores were low (M = 0.6), and remained low at 6 months (M= 0.7, Table 2). At 6 months, older age was significantly associated with lower scores (i.e., lesser concern regarding privacy). Highest level of education was also significant at 6 months, with those who completed less than a high school education having significantly higher scores (greater privacy concerns) than those who completed high school/GED (Table 3).

#### 3.2.4. Uncertainty Subscale

At 6 weeks, mean uncertainty scores were low (M = 2.9). There was a significant decrease in score at 6 months (M = 2.5) compared to 6 weeks (Table 2). Females had significantly higher uncertainty scores at 6 weeks, but this was no longer significant at 6 months. A significant association was found overall between years of education and uncertainty scores at both 6 weeks and 6 months. At six months, individuals with some college or a college degree/advanced degree had significantly lower scores (lesser uncertainty) compared to those whose highest level of education was completing high school/GED (Table 3). After adjusting the model for significant covariates (sex, education), time is still a significant predictor of the uncertain subscale score where there was a significant decrease in score at 6 months (Table 2). 

As can be appreciated on the forest plot (Figure 1), patient-participants reported low uncertainty, privacy, and negative emotions and moderate positive emotions, with variation between time points. 

### 3.3. PANAS

#### 3.3.1. Positive Affect 

At 6 weeks, mean PANAS positive affect score was 22.1, indicating midpoint levels of positive emotion. At 6 months, mean score (20.1) was significantly lower than at 6 weeks (Table 2). A family member having been told of the risk of the condition was significantly associated with positive affect at 6 weeks, with those who had a relative previously told having significantly higher scores compared to those who did not. Having a genetic counseling visit was significantly associated with positive affect at 6 weeks, with those who had a genetic counseling visit having higher scores compared to those who did not; this was no longer significant at 6 months (Table 4). After adjusting the model for significant covariates (having a genetic counseling visit), time is still a significant predictor of the positive subscale score where there was a significant decrease in value at 6 months (Table 2). 

#### 3.3.2. Negative Affect

At 6 weeks, mean PANAS negative affect score was 15.3, indicating low levels of negative emotion. At 6 months, mean score decreased (14.3), but was not significantly lower than at 6 weeks (Table 2). Having a genetic counseling visit was significantly associated with negative affect scores at 6 weeks and 6 months, with those who had genetic counseling having higher scores (greater negative affect) compared to those who did not at both time points. Female sex was significantly associated with higher scores at 6 weeks and 6 months, while older age was significantly associated with lower scores at both time points. Individuals with cardiac or miscellaneous results had significantly lower scores (less negative affect) compared to those with cancer results at 6 weeks; at 6 months only those with miscellaneous results had significantly lower scores than those with cancer results (Table 4).

Visual representation of the data (Figure 2) shows the changes in positive and negative affect scores over time.

#### 3.3.3. Discrete Emotions

Mean scores and change over time for discrete PANAS emotions are presented in Table 5. “Interested” was the highest scoring emotion at both time points. Mean scores of several positive emotions (interested, excited, strong, alert, inspired, determined, attentive, active) and negative emotions (distressed, upset, nervous, jittery, afraid) were significantly lower at 6 months than at 6 weeks, indicating a decrease in those emotions over time. None of the emotions had scores that were significantly higher at 6 months than at 6 weeks.

##### Decision Regret

Mean decision regret score was low at both 6 weeks (12.4) and 6 months (11.0), without a significant difference between time points (Table 2, Figure 3). Age at receiving results and sex were significantly associated with 6-week scores. Older individuals had higher decision regret scores, while females had lower scores. Education was significantly associated at both 6 weeks and 6 months. Individuals with some college or college/advanced degree had significantly lower scores (lesser regret) than those whose highest completed education level was high school/GED at 6 weeks, while those with less than a high school education had significantly higher scores than those with a high school level/GED at 6 months (Table 4). 

## 4. Discussion

The present study evaluates the psychological response of unselected individuals receiving clinically actionable results via a genomic screening program in a healthcare system. Overall, study participants reported moderate positive emotions and limited negative emotions as assessed by both FACToR and PANAS subscales across time. There were also low levels of uncertainty, privacy concern, and decision regret over time. 

This study’s results demonstrating low levels of negative emotional response in a large population genomic screening cohort are reassuring that receiving actionable genetic results in this setting may not cause psychological harm. Additionally, decision regret scores were low in our study, consistent with previous studies of biobank participants receiving actionable results [36,37,38]. A recent systemic literature review on return of secondary findings from genomic screening also suggested overall limited negative psychological impact, but concluded that variation in reporting of outcomes limited ability to draw conclusions and emphasized the need for more research [39]. However, while understanding outcomes from return of secondary findings in individuals sequenced for another clinical indication is valuable, it does not necessarily predict outcomes of population genomic screening programs given the different indications for testing. Thus, the current study provides encouraging insights on the psychological response of this population, which is particularly important in light of the growing number of population genomic screening programs. Participants with prior knowledge of a family member’s risk for the condition (about 1/3 of the study population) were significantly more likely to have greater positive feelings about the result on both the FACToR positive subscale and the PANAS positive affect, suggesting that these individuals may have anticipated such a result and, thus, were better able to process and adjust to the risk. We also saw a significant decrease in negative and uncertainty scores at 6 months on the FACToR scale. It appears that participants experienced some improved emotional adjustment over time, a key contribution to the understanding of psychological reactions to actionable results from genomic screening.

A broader, longitudinal understanding of emotional reactions to actionable genomic screening results provides insights into potential motivators and barriers to taking clinical action based on results. While it is reassuring that participants experienced limited negative emotions, strong emotions can be motivating [40] and may help patients respond appropriately to reduce disease risks (i.e., by taking appropriate next steps with recommended surveillance and management). Previous research by Rauscher and Hesse [28] on uncertainty and emotions demonstrates how strong emotional responses such as anxiety, interest, distress, pride, and nervousness can motivate family health history conversations. They found that discrete emotions might drive information seeking or avoidance behaviors differently, providing more insight than looking at positive and negative emotions broadly. Additionally, women with a pathogenic *BRCA1/2* variant have reported feeling high anxiety about their result and future cancer risks, which they described as motivating information seeking and adherence to surveillance guidelines [29,41]. As such, strong positive and/or negative emotional responses may not be psychologically harmful, but rather may prompt proactive medical decision making about genetic risks. Many of the PANAS positive emotions (e.g., interest, attentive, active, determined) are related to engagement and motivation [34]. While “interest” was the highest mean emotion in our cohort, participants’ scores for emotions such as active, attentive, and determined were all low (<3) and declined significantly from 6 weeks to 6 months post-disclosure. Our findings of relatively low (positive or negative) emotional response across time points and decreased discrete emotions related to engagement at 6 months post-disclosure highlight a potential challenge in how to motivate patients to take action related to medical management recommendations. It may be that concern for causing psychological harm has led to an over-correction in how genetic results are framed to reduce emotional response, which may in turn reduce potentially motivating emotions and engagement with recommended follow-up care. Future research should more thoroughly investigate what might be driving emotional responses to receiving an actionable result from genomic screening, including how discrete emotions impact medical decision making and adherence to management guidelines. 

Although our results demonstrate overall limited negative emotions and decision regret at both post-disclosure time points, more research is needed to understand why some individuals do experience these reactions, and how to best support and help them adapt to results. For example, in our population, individuals who were older, male, or reported lower educational attainment had higher decision regret scores, indicating that they may need additional resources as they make sense of their results and adjust to this information over time. Individuals with genetic results associated with increased risk for cardiovascular disease or a grouping of miscellaneous disease risks had lower levels of negative emotions than did those with genetic results associated with cancer risk. One hypothesis is that the population has more baseline awareness of, and therefore concerns about, cancer than about the included cardiac and miscellaneous conditions. Further evaluation of emotional response by specific genetic condition could also provide helpful insights, given the different risks associated with each condition. 

Differences in emotional response between certain subgroups raise the question of whether tailoring a program based on these characteristics might be beneficial. Changes have not yet been made to the MyCode program based on these findings, as more data are needed to understand why these differences exist. Further research on what is driving emotional response in different groups should guide future implementation strategies. Future modifications could include additional support, education, or tailored disclosure and genetic counseling practices for certain groups, either to help cope with negative feelings, or encourage health behaviors consistent with the result. Additionally, because some people do experience negative emotions, it is important that psychological support is available when needed in contexts where genomic screening results are disclosed. Future studies could also investigate the utility and practicality of routinely administering a tool to best identify those who might experience more negative emotions from genetic testing after genomic screening and provide support accordingly. Additionally, further study is needed to determine whether the consent process could be refined to better support individuals who might be anticipated to respond negatively to results disclosure. Future research should also examine how psychological reactions (FACToR), emotional response (PANAS), and decision regret impact family communication about the result. Previous research has found a common reported barrier to family communication and cascade testing for genetic conditions is difficulty sharing “bad news”, complex emotional responses from family members, and uncertainty about what information to share or difficulties recalling information accurately [42,43,44]. Characterizing psychological responses to receiving results from a genomic screening program can help researchers identify connections between these measures and behaviors such as medical management and family sharing. 

## 5. Limitations

It is important to interpret results and directions for future research while considering limitations of this study. Our population comes from a single healthcare system, and the study population was not ethnically or racially diverse, meaning it may not be representative of more diverse populations found at other US-based healthcare systems. Further, while the longitudinal nature of these data is a strength, surveying this population represents retention limitations. The response rate for this study was approximately 29%, which additionally limits the generalizability as it is possible that we would have seen different findings (more negative or more positive emotions) if there had been a higher response rate. It could be that individuals who felt the strongest (either positively or negatively) responded to both surveys and we may be missing other viewpoints in the data. However, the rigor of Geisinger’s Survey Research and Recruitment Core’s approach to surveying participants may have mitigated selection bias issues.

## 6. Conclusions

Despite these limitations, our findings can help inform the clinical utility of genomic screening in unselected healthcare system populations. Individuals appear to experience limited psychological harm that may wane over time. This, coupled with other benefits of genomic screening, adds support to the clinical utility of population genomic screening programs. Overall low levels of discrete positive and negative emotions, however, raise questions about how to best elicit an emotional response that motivates behavior change without causing psychological harm.

## Figures and Tables

**Figure 1 jpm-12-01943-f001:**
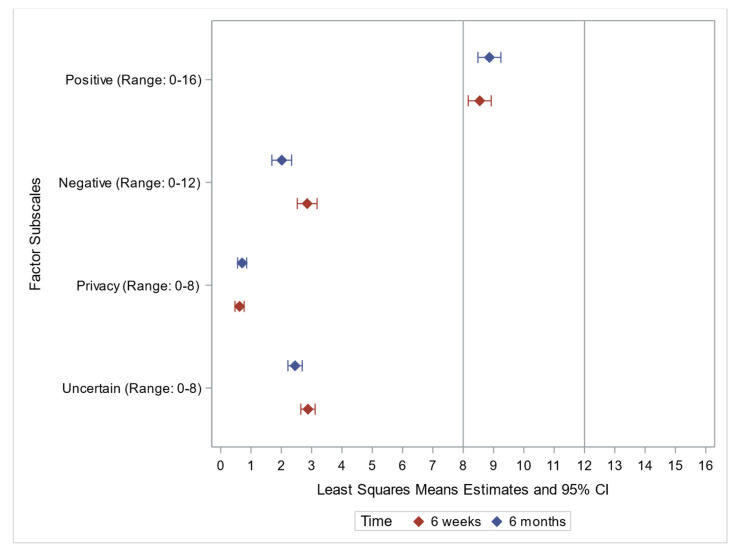
FACToR Scale Forest Plot.

**Figure 2 jpm-12-01943-f002:**
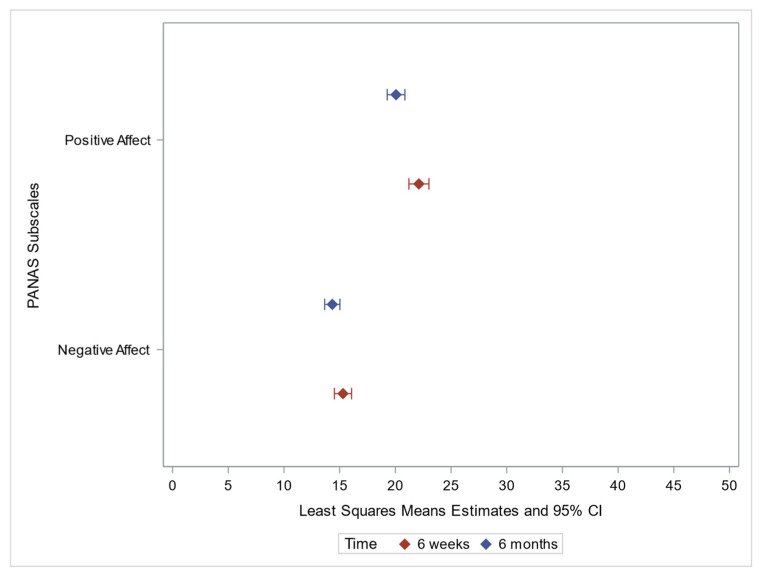
PANAS subscales Forest Plot.

**Figure 3 jpm-12-01943-f003:**
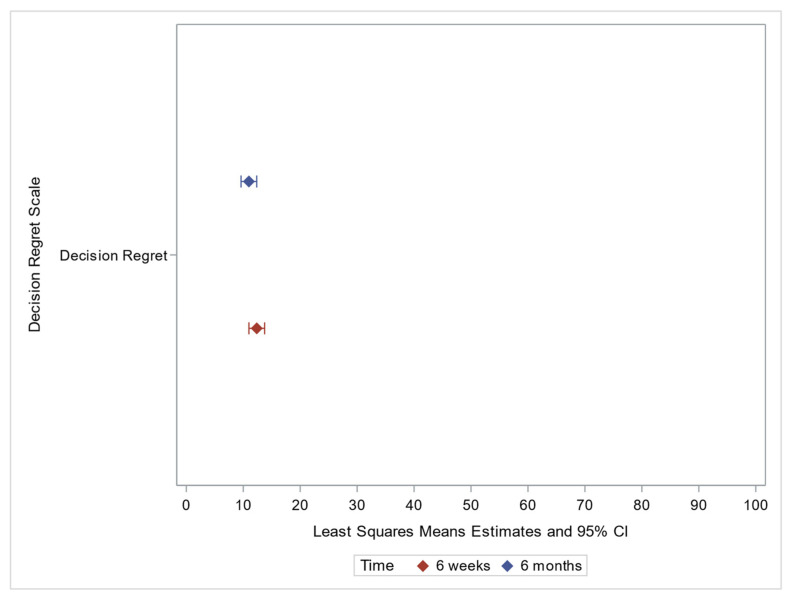
Decision Regret Factor Plot.

**Table 1 jpm-12-01943-t001:** Sample Characteristics.

	Total (n = 354)
Age at result receipt	
Mean (SD)	57.6 (15.72)
Median (IQR)	60.6 (47.1, 69.6)
Range	21.7, 89.5
Sex, n (%)	
Female	221 (62.6%)
Male	132 (37.4%)
Missing	1
Race, n (%)	
White	310 (87.6%)
Asian	2 (0.6%)
Black Or African American	2 (0.6%)
Native Hawaiian or Other Pacific Islander	1 (0.3%)
Unknown	39 (11.0%)
Ethnicity, n (%)	
Hispanic or Latino	5 (1.4%)
Not Hispanic or Latino	310 (87.6%)
Unknown	39 (11.0%)
What is your current marital status, n (%)	
Married/Living with partner	235 (66.8%)
Divorced/Separated	43 (12.2%)
Widowed	34 (9.7%)
Never Married	40 (11.4%)
Missing	2
What is the highest grade or year of school you completed, n (%)	
Less than high school	15 (4.3%)
Completed high school/GED	90 (25.6%)
Some College	119 (33.8%)
College grad or Advanced degree	128 (36.4%)
Missing	2
What is your annual HOUSEHOLD income from all sources, n (%)	
Less than $49,999	103 (29.1%)
$50,000–$99,999	104 (29.4%)
$100,000 or more	69 (19.5%)
I prefer not to answer	78 (22.0%)
Have you ever been told that you were at risk for a genetic condition before getting a call from the My Code team? (6 weeks only) n (%)	
Yes	61 (21.9%)
No	217 (78.1%)
Missing	76
Has anyone in your family been told of a risk for THIS genetic condition? (6 weeks only) n (%)	
Yes	95 (34.2%)
No	176 (63.3%)
Don’t know	7 (2.5%)
Missing	76
How would you say your HEALTH is? (6 months), n (%)	
Mean (SD)	2.8 (1.00)
Median (IQR)	3.0 (2.0, 3.0)
How would you say your QUALITY OF LIFE is? (6 months), n (%)	
Mean (SD)	2.3 (0.97)
Median (IQR)	2.0 (2.0, 3.0)
Gene group, n (%)	
Cancer	157 (44.4%)
Cardiac	116 (32.8%)
Miscellaneous phenotype (*HFE, SMAD4, FBN1, TGFBR1, TGFBR2, SMAD3, OTC, PTEN, TSC1, TSC2, COL3A1, ATP7B, RYR1, GLA*)	81 (22.9%)
Completed GC visits, n (%)	182 (51.4%)
Before T1	166 (91.2%)
Between T1-T2	16 (8.8%)

**Table 2 jpm-12-01943-t002:** Descriptive Statistics and Generalized linear models assessing relationship between time and outcomes.

	6 Weeks (n = 354)	6 Months (n = 354)	*p*-Value	Adjusted *p*-Value
FACToR scales				
Positive subscale (0–16)				
Mean (SD)	8.5 (3.56)	8.9 (3.70)	0.2422	
Range	0.0, 16.0	0.0, 16.0		
Negative subscale (0–12)				
Mean (SD)	2.9 (3.22)	2.0 (3.02)	0.0004	(Estimate (SE): −0.82 (0.140), *p*-value < 0.0001) *
Range	0.0, 12.0	0.0, 12.0		
Privacy subscale (0–8)				
Mean (SD)	0.6 (1.34)	0.7 (1.52)	0.4315	
Range	0.0, 8.0	0.0, 8.0		
Uncertain subscale (0–8)				
Mean (SD)	2.9 (2.25)	2.5 (2.27)	0.0126	(Estimate (SE): −0.42 (0.136), *p*-value = 0.0020) **
Range	0.0, 8.0	0.0, 8.0		
PANAS Scale				
Positive Affect (0–50)				
Mean (SD)	22.1 (7.91)	20.1 (7.38)	0.0008	(Estimate (SE): −2.15 (0.449), *p*-value < 0.0001) ***
Range	10.0, 49.0	10.0, 44.0		
Negative Affect (0–50)				
Mean (SD)	15.3 (6.72)	14.3 (6.42)	0.0681	
Range	10.0, 47.0	6.0, 47.0		
Decision Regret Scale (0–100)				
Mean (SD)	12.4 (12.84)	11.0 (13.59)	0.1640	
Range	0.0, 90.0	0.0, 75.0		

* time is still significant after adjusting for GC visit, sex, gene group, age. ** time is still significant after adjusting for sex and education. *** time is still significant after adjusting for GC visit and prior family knowledge of condition.

**Table 3 jpm-12-01943-t003:** General linear model estimates for the FACToR subscales at 6 weeks and 6 months.

	Six Weeks				Six Months			
	Positive	Negative	Privacy	Uncertain	Positive	Negative	Privacy	Uncertain
	*p*-Value Est (SE)	*p*-Value Est (SE)	*p*-Value Est (SE)	*p*-Value Est (SE)	*p*-Value Est (SE)	*p*-Value Est (SE)	*p*-Value Est (SE)	*p*-Value Est (SE)
**GC visit**	0.1865	0.0092	0.0827	0.2183	0.5406	0.0702	0.0636	0.3974
Yes	−0.50 (0.378)	0.89 (0.340)	0.25 (0.142)	0.30 (0.239)	−0.24 (0.394)	0.58 (0.320)	0.30 (0.161)	−0.20 (0.241)
No (ref)	*-*	*-*	*-*	*-*	*-*	*-*		*-*
**Age at result receipt**	<0.0001	0.0855	0.1555	0.9987	0.0004	0.0028	0.0015	0.4419
	0.05 (0.012)	−0.02 (0.011)	−0.01 (0.004)	−0.00001 (0.007)	0.04 (0.012)	−0.03 (0.010)	−0.02 (0.005)	−0.006 (0.008)
**Sex**	0.0007	0.0061	0.1348	0.0235	0.0222	0.0030	0.0551	0.3098
Female	−1.32 (0.386)	0.97 (0.351)	0.22 (0.147)	0.56 (0.245)	−0.93 (0.405)	0.98 (0.329)	0.32 (0.166)	0.25 (0.250)
Male (ref)	-	-	-	-	-	-	-	-
**Highest Education**	0.9173	0.5677	0.3485	0.0141	0.1780	0.1598	0.0070	0.0004
Less than high school	−0.16 (0.997)	0.76 (0.902)	−0.10 (0.375)	0.0642 1.16 (0.622)	−1.42 (1.029)	1.58 (0.837)	0.0367 0.88 (0.418)	0.9141 0.07 (0.618)
Completed high school/GED (ref)	-	-	-	-	-	-	-	-
Some College	−0.08 (0.499)	−0.38 (0.452)	−0.32 (0.188)	0.9508 −0.02 (0.312)	−0.07 (0.515)	−0.18 (0.419)	0.1120 −0.33 (0.210)	0.0011 −1.02 (0.309)
College grad/Advanced degree	0.21 (0.492)	−0.23 (0.445)	−0.26 (0.185)	0.0582 −0.58 (0.307)	0.57 (0.508)	−0.25 (0.413)	0.0590 −0.39 (0.206)	0.0002 −1.16 (0.305)
**Has anyone in your family been told of a risk for THIS genetic condition?**	0.0016	0.8142	0.5661	0.7981	N/A	N/A	N/A	N/A
Yes	0.0003 −1.60 (0.443)	0.26 (0.414)	0.18 (0.173)	0.01 (0.294)	N/A	N/A	N/A	N/A
No (ref)	-	-	-	-	N/A	N/A	N/A	N/A
Don’t know	0.8292−0.29 (1.342)	0.02 (1.254)	−0.01 (0.524)	−0.59 (0.889)	N/A	N/A	N/A	N/A
**Gene group**	0.0120	0.0044	0.1044	0.4717	0.5772	0.0001	0.5358	0.1417
Cancer (ref)	-	-	-	-	-	-	-	-
Cardio	0.0358	0.0945				0.0031		
	0.91 (0.431)	−0.65 (0.390)	−0.24 (0.164)	0.23 (0.276)	0.16 (0.453)	−1.08 (0.362)	−0.04 (0.186)	−0.36 (0.277)
Miscellaneous	0.2451	0.0011				<0.0001		
	−0.56 (0.482)	−1.43 (0.435)	−0.36 (0.183)	−0.16 (0.308)	−0.40 (0.507)	−1.60 (0.404)	−0.23 (0.208)	−0.58 (0.309)

**Table 4 jpm-12-01943-t004:** General linear model estimates for the Decision Regret, Positive and Negative Affect (PANAS) scales at 6 weeks and 6 months.

	Six Weeks			Six Months		
	PANAS Positive Affect	PANAS Negative Affect	Decision Regret	PANAS Positive Affect	PANAS Negative Affect	Decision Regret
	*p*-Value Est (SE)	*p*-Value Est (SE)	*p*-Value Est (SE)	*p*-Value Est (SE)	*p*-Value Est (SE)	*p*-Value Est (SE)
**GC visit**	0.0107	0.0154	0.1941	0.3023	0.0499	0.0552
Yes	2.52 (0.983)	2.04 (0.836)	−1.77 (1.364)	0.81 (0.785)	1.34 (0.680)	−2.77 (1.440)
No (ref)	-	-	-	-	-	-
**Age at result receipt**	0.1554 −0.04 (0.030)	0.0440 −0.05 (0.025)	0.0223 0.10 (0.043)	0.6798 −0.01 (0.025)	0.0033 −0.06 (0.022)	0.2286 0.06 (0.046)
**Sex**	0.1880	0.0005	0.0032	0.1188	0.0003	0.1051
Female	1.28 (0.966)	2.88 (0.811)	−4.14 (1.397)	1.27 (0.811)	2.53 (0.695)	−2.42 (1.492)
Male (ref)	-	-	-	-	-	-
**Highest Education**	0.9281	0.3638	0.0039	0.4707	0.2560	0.0255
Less than high school	−1.00 (2.810)	0.79 (2.389)	0.1774 4.78 (3.535)	−0.57 (2.046)	1.91 (1.780)	0.0130 9.39 (3.762)
Completed high school/GED (ref)	-	-	-	-	-	-
Some College	−0.18 (1.264)	−1.32 (1.074)	0.0106 −4.55 (1.771)	−0.27 (1.025)	−0.82 (0.892)	0.5876 −1.02 (1.884)
College grad/Advanced degree	−0.71 (1.231)	−1.60 (1.046)	0.0164 −4.21 (1.744)	−1.44 (1.009)	−1.11 (0.878)	0.3523 −1.73 (1.856)
**Has anyone in your family been told of a risk for THIS genetic condition?**	0.0063	0.2402	0.3121	N/A	N/A	N/A
Yes	0.0047 2.83 (0.992)	1.30 (0.854)	−2.09 (1.679)	N/A	N/A	N/A
No (ref)	-	-	-	N/A	N/A	N/A
Don’t know	0.0724 5.42 (3.004)	−1.46 (2.584)	3.73 (5.082)	N/A	N/A	N/A
**Gene group**	0.1747	0.0207	0.1953	0.2677	0.0141	0.0935
Cancer (ref)	-	-	-	-	-	-
Cardio	−1.89 (1.080)	0.0289 −2.00 (0.910)	−1.97 (1.569)	−1.18 (0.903)	0.1077 −1.26 (0.779)	−0.64 (1.658)
Miscellaneous	−1.55 (1.252)	0.0169 −2.54 (1.055)	−2.95 (1.753)	−1.40 (1.009)	0.0042 −2.51 (0.870)	−3.95 (1.852)

**Table 5 jpm-12-01943-t005:** Discrete PANAS Emotions and Change Over Time.

PANAS Emotion	6 weeks Survey, Mean (SD) (n = 278)	6 months Survey, Mean (SD) (n = 278)	*p*-Value
Positive			
Interested	3.6 (1.13)	3.3 (1.21)	<0.0001
Excited	1.5 (1.00)	1.4 (0.87)	0.0456
Strong	2.2 (1.29)	1.9 (1.31)	0.0254
Enthusiastic	1.5 (1.00)	1.4 (0.95)	0.2176
Proud	1.4 (0.94)	1.3 (0.79)	0.0920
Alert	2.6 (1.25)	2.3 (1.35)	0.0003
Inspired	1.8 (1.18)	1.6 (1.11)	0.0232
Determined	2.4 (1.29)	2.2 (1.38)	0.0212
Attentive	2.7 (1.31)	2.5 (1.35)	0.0169
Active	2.4 (1.36)	2.0 (1.17)	<0.0001
Negative			
Distressed	1.8 (1.10)	1.6 (1.04)	0.0179
Upset	1.8 (1.12)	1.6 (1.02)	0.0035
Guilty	1.2 (0.72)	1.2 (0.76)	0.4359
Scared	1.8 (1.05)	1.7 (1.07)	0.4433
Hostile	1.0 (0.36)	1.1 (0.37)	0.4237
Irritable	1.3 (0.71)	1.3 (0.80)	0.7841
Ashamed	1.1 (0.51)	1.1 (0.49)	0.8116
Nervous	2.0 (1.14)	1.8 (1.08)	0.0029
Jittery	1.5 (0.89)	1.3 (0.78)	0.0050
Afraid	1.8 (1.08)	1.6 (1.03)	0.0106

## Data Availability

The data presented in this study are available on request from the corresponding author. The data are not publicly available due to the potential to identify participants.

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
