# Peer review of "Investigating Psychological Impact after Receiving Genetic Risk Results—A Survey of Participants in a Population Genomic Screening Program"

_jpm, 2022, doi:10.3390/jpm12121943_

Round 1

Reviewer 1 Report

This is a well-written paper and well-conducted study, of high quality. It makes an important contribution to the field and the authors should be commended.

Comments.

1. Of particular interest to me were the sub-group findings, e.g. that responses to genetic results differed between men vs women, in older people, by income group etc, and by condition (cancer vs cardiac result). I wondered whether these differences would be enough to warrant the customization of a population genomic screening program to certain subgroups? I further wondered whether the MyCode program has itself changed or modified (or is planning to) any of its return of results or genetic counselling processes based on these findings (e.g. begin to customize your processes to accordingly different subgroups, health conditions etc)? If so, mention of these changes in the manuscript (to demonstrate the practical, real-world benefits/application of the study) would strengthen the paper. Alternatively, if no changes to the MyCode processes have been made so far on account of these findings, then what additional evidence is required to begin start customizing the genomic screening/return of results to different subgroups?

2. Response rate. I think the response rate needs to be mentioned in the abstract (e.g. “Of the 1208 individuals who received both the T1 and T2 surveys, 354 individuals 157 completed both surveys, for a response rate of 29.3%.”) In following, the responding cohort is likely to be subject to some fairly strong volunteer/responder bias, which maybe needs to be mentioned more in the Limitations section (at present it is a little under-stated…)

3. Negative control group. I noticed there was no negative a control group (e.g. people from the MyCode network who underwent genetic testing but did not receive a high-risk result). Was this a deliberate/necessary decision re the study design (e.g. due to the scales used)? Would a negative control group have added any benefit to the study?

4. A fairly high portion of participants either had previously been told they were at risk of the condition (21.9%) or had a family member who had been told (34.2%). Could this have affected their psychological response to the genetic results? (e.g. those with prior knowledge of the condition responded to the result more favourably?) Perhaps this point could be elaborated in the Discussion more. I think a key question in the field right now is whether people who undertake clinical genetic testing (eg ascertained with known family history or family variant) would respond differently to genetic risk information, compared with people tested from the general population (otherwise healthy, with no family history or prior knowledge of the disease). If your study can shed any light on that question, I think it would be of great interest.

5. The finding that Cardiac results had higher scores (greater functional impairment) than cancer is interesting. This could be elaborated in the Discussion a bit more. Also, I wondered if there any discernible difference between FH (familial hypercholesteremia) results vs other cardiac conditions (eg those associated with sudden cardiac death)?

Minor

1. Intro- “Population genomic screening can identify individuals at risk for disease who might not otherwise be ascertained by clinical criteria-based testing [1–4].

2. Probably not necessary to show income data in table 1 (maybe condense into 3 categories instead of 10)

3. Maybe not necessary to show Median (IQR) rows in Table 2, in addition to Mean(SD)

4. There are some missing recent references on population genomic screening, esp PMID: 35267197 that should be cited

Reviewer 2 Report

The authors investigate the psychological impact after receiving genetic risk results in a population undergoing genetic screening program. The addressed topic is of high clinical and social interest as the genetic testing is becoming even more cheaper and available for the population, making on the same time the related ethical concerns a hot topic as well. The perspective, that is offered from this study is noteworthy. A possible further development of this study, which I will not recommend as revision/change of this paper, should be the possibility to evaluate by similar psychological tools groups of people who have not yet taken part to such genetic testing screening program, and could be predicted for them, to possibly identify people to be excluded (or, at least, for which the opportunity of their inclusion should be carefully evaluated prior undergoing such programs), thus avoiding unexpected negative psychological burden in the absence of a relevant advantages driven by the genetic results. Such a perspective could be briefly summarized in the discussion as a proposal of further study.

Author Response

Reviewer #2

The authors investigate the psychological impact after receiving genetic risk results in a population undergoing genetic screening program. The addressed topic is of high clinical and social interest as the genetic testing is becoming even more cheaper and available for the population, making on the same time the related ethical concerns a hot topic as well. The perspective, that is offered from this study is noteworthy. A possible further development of this study, which I will not recommend as revision/change of this paper, should be the possibility to evaluate by similar psychological tools groups of people who have not yet taken part to such genetic testing screening program, and could be predicted for them, to possibly identify people to be excluded (or, at least, for which the opportunity of their inclusion should be carefully evaluated prior undergoing such programs), thus avoiding unexpected negative psychological burden in the absence of a relevant advantages driven by the genetic results. Such a perspective could be briefly summarized in the discussion as a proposal of further study.

Response: Thank you for the recognition of the manuscript topic as being of high clinical and social interest. Although the current study shows that the vast majority of people do not have significant negative psychological impact, we appreciate the reviewer’s thoughts about this future direction. Further study is needed to determine whether the consent process could be refined to better support individuals who might be anticipated to respond negatively to result disclosure, or to identify a specific group of people at higher risk for negative outcomes prior to testing which would help us to develop additional support for this small subset of individuals. We incorporated this as a future direction of study in the discussion.